# Tricyano-Methylene-Pyridine Based High-Performance Aggregation-Induced Emission Photosensitizer for Imaging and Photodynamic Therapy

**DOI:** 10.3390/molecules27227981

**Published:** 2022-11-17

**Authors:** Xupeng Wu, Zhirong Zhu, Zhenxing Liu, Xiangyu Li, Tijian Zhou, Xiaolei Zhao, Yuwei Wang, Yiqi Shi, Qianqian Yu, Wei-Hong Zhu, Qi Wang

**Affiliations:** 1Shanghai Key Laboratory of Functional Materials Chemistry, Key Laboratory for Advanced Materials and Institute of Fine Chemicals, Joint International Research Laboratory of Precision Chemistry and Molecular Engineering, Feringa Nobel Prize Scientist Joint Research Center, Frontiers Science Center for Materiobiology and Dynamic Chemistry, School of Chemistry and Molecular Engineering, East China University of Science and Technology, Shanghai 200237, China; 2Research Center of Resources and Environment, School of Chemical Engineering and Materials, Changzhou Institute of Technology, Changzhou 213032, China

**Keywords:** aggregation-induced emission, photosensitizer, ROS generation, bioimaging

## Abstract

Photosensitizers equipped with high reactive oxygen species (ROS) generation capability and bright emission are essential for accurate tumor imaging and precise photodynamic therapy (PDT). However, traditional aggregation-caused quenching (ACQ) photosensitizers cannot simultaneously produce desirable ROS and bright fluorescence, resulting in poor image-guided therapy effect. Herein, we report an aggregation-induced emission (AIE) photosensitizer TCM-Ph with a strong donor–π–acceptor (D–π–A) structure, which greatly separates the HOMO–LUMO distribution and reduces the Δ*E*_ST_, thereby increasing the number of triplet excitons and producing more ROS. The AIE photosensitizer TCM-Ph has bright near-infrared emission, as well as a higher ROS generation capacity than the commercial photosensitizers Bengal Rose (RB) and Chlorine e6 (Ce6), and can effectively eliminate cancer cells under image guidance. Therefore, the AIE photosensitizer TCM-Ph has great potential to replace the commercial photosensitizers.

## 1. Introduction

Photodynamic therapy (PDT) is a noninvasive therapeutic method that can kill cancer cells effectively. Photosensitizers equipped simultaneously with high reactive oxygen species generation (ROS), bright near-infrared (NIR) fluorescence, and excellent photostability are crucial for obtaining optimal PDT performance [1,2]. However, traditional photosensitizers, such as Bengal Rose (RB) and Chlorine e6 (Ce6), often suffer from low ROS generation efficiency, fluorescence quenching caused by intermolecular π–π stacking [3], and photobleaching effect, which are difficult to be used for imaging-guided PDT [4,5,6]. In contrast, aggregation-induced emission luminogens (AIEgens) with high photostability and bright emission in the aggregated state, capable of reducing nonradiative leaps, are expected to overcome the dilemma of commercial photosensitizers [7,8]. Therefore, the design of a new generation of AIE photosensitizers is of great significance for image-guided PDT [9,10].

With this in mind, the AIE photosensitizer TCM-Ph with bright fluorescence and high ROS generation is rationally designed based on our previously reported AIE building block tricyano-methylene-pyridine (TCM) (Figure 1A). The π-bridge thiophene group and strong electron donor carbazole could induce a strong intramolecular charge transfer (ICT) [11], increase the Stokes shift, and extend the emission wavelength to the NIR region, thus reducing spontaneous interference from biological tissue and enabling accurate bioimaging [12]. The three strong electron-withdrawing groups cyano endow TCM-Ph with a strong donor–π–acceptor structure, which could promote the charge separation of excited states, thus helping to reduce the energy gap between the excited singlet state and the excited triplet state (Δ*E*_ST_). The introduction of phenyl further promotes the separation of HOMO–LUMO and effectively reduces Δ*E*_ST_ [13,14,15], which is conductive to enhancing the intersystem crossing process (ISC) for efficient ROS generation [16].

TCM-Ph exhibits a broad absorption spectrum, which is beneficial for matching and utilizing the white-light spectrum. Moreover, the TCM-Ph exhibits strong fluorescence intensity and better ROS generation efficiency than RB and Ce6, which makes it have the potential of image-guided PDT [17]. The encapsulation of amphiphilic polymers DSPE-PEG_2000_ and DSPE-PEG_2000_-FA enhances the cell penetration of TCM-Ph (Figure 1B), and does not affect the ROS production efficiency, making it an ideal choice for efficient tumor-targeted therapy.

## 2. Results and Discussions

### 2.1. Developing AIE-Active Luminogens as Photosensitizer

Photosensitizers with bright emission, high ROS generation, as well as excellent photostability are the key for image-guided PDT. In this work, we designed an AIE photosensitizer TCM-Ph for image-guided PDT based on the previously reported AIE building block TCM [18,19,20,21,22]. As illustrated in Figure 1A, the thiophene and carbazole moieties were first introduced into two sites of TCM, which can not only extend the emission wavelength to the NIR region for precise bioimaging, but also form a strong D–π–A structure to separate the HOMO–LUMO distribution, thus facilitating reducing Δ*E*_ST_. The phenyl group further reduced Δ*E*_ST_, thereby enhancing ISC process and favoring the ROS generation [17]. The structures of TCM-based photosensitizers TCM-Et and TCM-Ph were confirmed by nuclear magnetic resonance (NMR) spectra and mass spectrum.

### 2.2. Photophysical Properties of Photosensitizer TCM-Ph

The photophysical properties of both TCM-Et and TCM-Ph were evaluated in water/tetrahydrofuran (THF) mixtures with different water fractions. Both AIE photosensitizers had an absorption range of 400–600 nm with a maximum peak around 510 nm (Figure 2A,E), and the broad absorption was favorable for ROS generation under white-light irradiation [14,23]. Meanwhile, TCM-Et (Figure 2B,C) and TCM-Ph (Figure 2F,G) exhibited typical AIE properties in that they dissolved well in THF with no fluorescent signal and gradually aggregated with an increased fluorescence signal as the water fraction increased to 80%. Moreover, the maximum emission wavelength of TCM-Et was red-shifted from 591 nm to 632 nm with increasing water content, while the wavelength of TCM-Ph was red-shifted from 593 nm to 641 nm. The long emission wavelength could effectively reduce the interference of background fluorescence. The aggregated states of TCM-Et and TCM-Ph were further demonstrated by dynamic light scattering (DLS) (Figure 2D,H) and transmission electron microscope (TEM) (Appendix A), such that both compounds exhibited a spherical shape in 99% water with the size of 70 nm. In addition, the compounds TCM-Et and TCM-Ph exhibited good fluorescence intensities with solid-state absolute fluorescence quantum yields of 2.5% and 1.3% (Appendix A), respectively, offering the possibility of bioimaging.

### 2.3. High ROS Generation of TCM-Ph

The photosensitivity of these photosensitizers was evaluated in comparison to commercially available photosensitizers (Ce6 and RB). First, 2′,7′-dichlorodihydrofluorescein (DCFH) was used as an indicator to evaluate total ROS generation (Appendix A) [24,25]. In the presence of Ce6 and RB, the fluorescence of DCFH at 525 nm increased 5.9 and 14.5 times after 3.5 min of light exposure, respectively (Figure 3A), while the fluorescence of DCFH increased 25.6 times and 29.0 times after being treated with TCM-Et and TCM-Ph, respectively, indicating that both of these compounds could generate more ROS than commercial RB and Ce6. Then, the ^1^O_2_ generation of TCM-Ph was evaluated using 9,10-anthracenediyl-bis(methylene) dimalonic acid (ABDA) as an indicator [24,26]. As shown in Figure 3B, the absorbance of ABDA was constant in the absence of photosensitizers. In contrast, in the presence of photosensitizers Ce6, TCM-Et, RB and TCM-Ph, the absorbance of ABDA decreased to 22.2%, 43.8%, 56.6% and 64.1%, respectively, after 4 min of light exposure (Appendix A), and the decomposition rate constants decreased in the order of TCM-Ph (0.2542), RB (0.2092), TCM-Et (0.1422) and Ce6 (0.0602 min^−1^), indicating that the ^1^O_2_ generation rate of TCM-Ph was the highest among these four photosensitizers (Figure 3C). In addition, TCM-Ph may produce superoxide radical (O_2_^−·^) (Appendix A). These results indicated that TCM-Ph was an excellent potential commercial photosensitizer candidate. Time-dependent density functional theory (TD-DFT) calculations and low temperature spectra were then performed to illustrate the ROS generation mechanism of TCM-Ph [27,28]. As shown in the calculation results, the highest occupied molecular orbital (HOMO) and the lowest unoccupied molecular orbital (LUMO) of TCM-Et partially overlapped in the TCM core, while the HOMO electron cloud of TCM-Ph was mainly distributed in the thiophene-carbazole unit, which was separated well from the LUMO electron cloud (Figure 3D) [9,11,14,29]. The energy gap between the excited singlet and the excited triplet of TCM-Ph (Δ*E*_S1-T1_, 0.68 eV) was much smaller than that of TCM-Et (Δ*E*_S1-T1_, 0.71 eV) (Figure 3D, Appendix A). The low-temperature fluorescence and phosphorescence spectra (Appendix A) further estimated the energy gap between the S1 and T1 states of the photosensitizers [30], and obtained similar results to the calculated ones, such that the Δ*E*_S1-T1_ value of TCM-Ph (0.66 eV) was smaller than that of TCM-Et (0.87 eV) (Figure 3E), which facilitated the singlet-triplet ISC process and the efficient generation of ROS [31].

### 2.4. TCM-Ph NPs for Efficient Cell Penetrating and Photodynamic Therapy

As evaluated, TCM-Ph was the most promising alternative to commercial photosensitizers due to the highest ROS generation. Therefore, amphiphilic polymer DSPE-PEG_2000_ and DSPE-PEG_2000_-FA (containing folic acid groups) were used to encapsulate the photosensitizer TCM-Ph, and the obtained TCM-Ph nanoparticles (TCM-Ph NPs, TCM-Ph/DSPE-mPEG_2000_/DSPE-PEG_2000_-FA, m/m/m, 1/4.8/1.2) had improved water dispersibility for biological applications. The size distribution of TCM-Ph NPs was around 109 nm with spherical morphology (Figure 4A and Appendix A), which showed similar absorption (500 nm, Figure 4B), maximum emission (636 nm, Figure 4B), and high photosensitivity (Figure 4C and Appendix A) as TCM-Ph. Then, the cell-penetrating ability of TCM-Ph NPs was evaluated through incubating with HeLa cells. After incubating for 6 h, the unpacked TCM-Ph could not be taken up by tumor cells (Appendix A), while the TCM-Ph NPs treatment cells exhibited efficient cell penetration with strong intracellular fluorescence (Figure 4D,E).

Furthermore, the intracellular ROS generation ability of TCM-Ph NPs was evaluated by DCFH-DA [32,33]. First, HeLa cells were incubated with TCM-Ph NPs for 6 h, and then treated with DCFH-DA (20 μM) for 20 min. After illuminating with white light for 6 min, images of HeLa cells were collected by a Confocal Laser Scanning Microscope (CLSM) (Malvern Instruments, Worcestershire, UK). As shown in Figure 4F, no fluorescence was observed in the control groups (photosensitizer alone or lighting alone), while bright green fluorescence was observed in the photodynamic treatment group (photosensitizer + light), indicating that TCM-Ph NPs could increase intracellular ROS level upon light irradiation. All of these results demonstrated the efficient intracellular ROS generation ability of TCM-Ph NPs.

Subsequently, the performance of TCM-Ph NPs as a photosensitizer for photo ablation of cancer cells was examined by MTT assay using HeLa cells [34,35]. The results showed that, in dark condition, TCM-Ph NPs exhibited low cytotoxicity (Figure 4G). In contrast, under light condition, the nanoparticles showed increased cytotoxicity with increasing concentration. Specifically, after 10 min of irradiation, cell viability was 83.3% at a photosensitizer concentration of 1 μg/mL, while it decreased to 43.6% when the concentration was increased to 20 μg/mL. In addition, cytotoxicity was enhanced with increasing irradiation time, for example, cell viability was 43.6% at a photosensitizer concentration of 20 μg/mL after 10 min of irradiation, and cell viability decreased to 6.9% after 20 min of irradiation (Figure 4G). These results demonstrated that TCM-Ph NPs could play a role as a significant photosensitizer with efficient tumor cells killing ability.

## 3. Conclusions

To develop a high-performance photosensitizer with bright emission and good ROS generation for image-guided anti-tumor therapy, we rationally designed a NIR AIE photosensitizer TCM-Ph by introducing the electron-donating carbazole unit and π-bridged thiophene into AIE building block TCM to extend the wavelength, and further introducing phenyl into TCM to improve the ISC efficiency to generate ROS. As expected, TCM-Ph exhibited higher ROS generation efficiency than the commercial photosensitizers RB and Ce6. More importantly, after being encapsulated with amphiphilic polymers DSPE-PEG and DSPE-PEG-FA, the acquired TCM-Ph NPs exhibited efficient cell-penetrating ability, and high intracellular ROS generation for tumor cells eliminating, providing an alternative candidate to commercial photosensitizers.

## Figures and Tables

**Figure 1 molecules-27-07981-f001:**
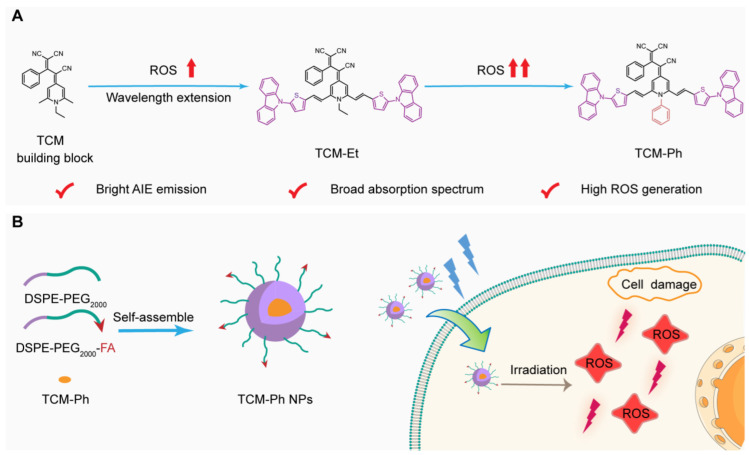
Design of the high-performance aggregation-induced emission photosensitizer TCM-Ph for imaging and photodynamic therapy. (**A**) Rational design of the AIE photosensitizer TCM-Ph; (**B**) TCM-Ph nanoparticles (TCM-Ph NPs) could image cancer cells and produce ROS upon light irradiation to kill cancer cells.

**Figure 2 molecules-27-07981-f002:**
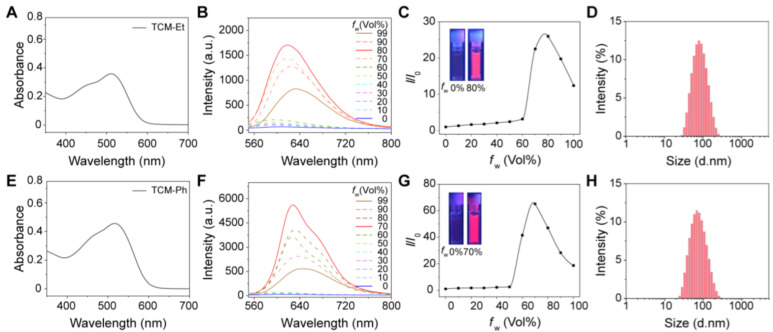
Photophysical properties of TCM-Et and TCM-Ph. (**A**) Absorption spectra of TCM-Et in 99% water; (**B**) Fluorescence spectra and (**C**) *I*/*I*_0_ plots of TCM-Et in THF/water mixtures with different water fractions (*f*_w_), *λ*_ex_ = 511 nm. Inset: photographs of TCM-Et in 0% and 80% water under UV lamp (365 nm). *I*_0_ is the maximum fluorescence intensity of TCM-Et in THF. (**D**) Size distribution of TCM-Et in 99% water; (**E**) Absorption spectra of TCM-Ph in 99% water; (**F**) Fluorescence spectra and (**G**) *I*/*I*_0_ plots of TCM-Ph in THF/water mixtures with different *f*_w_, *λ*_ex_ = 510 nm. Inset: photographs of TCM-Ph in 0% and 70% water under UV lamp (365 nm). *I*_0_ is the maximum fluorescence intensity of TCM-Ph in THF. (**H**) Size distribution of TCM-Ph in 99% water.

**Figure 3 molecules-27-07981-f003:**
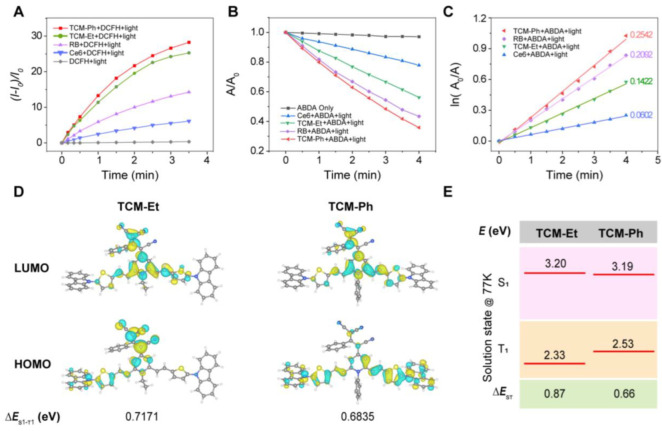
High ROS generation of TCM-Ph. (**A**) Light-induced total ROS generation of Ce6, RB, TCM-Et and TCM-Ph (10 μM) with DCFH (40 μM) as indicator. The plot of relative PL intensity (*I*-*I*_0_)/*I*_0_ at 525 nm versus the different irradiation times, where *I*_0_ is the fluorescence value of the mixture at 525 nm before illumination, and *I* is the fluorescence value of the mixture at 525 nm after illumination; excitation wavelength is 488 nm. (**B**) Light-induced ^1^O_2_ generation of Ce6, RB, TCM-Et and TCM-Ph (10 μM), evaluated by ABDA (50 μM). The relationship between *A/A*_0_ of ABDA at 378 nm versus the different irradiation times. (**C**) The relationship between ln(*A*_0_*/A*) and the light irradiation times, where *A*_0_ is the absorbance of ABDA at 378 nm before light irradiation, and *A* is the absorbance of ABDA at 378 nm after light irradiation. (**D**) Molecular orbital amplitude plots of HOMO and LUMO. (**E**) The energy levels of TCM-Et and TCM-Ph. The well vibration-resolved profiles allow accurate energy level abstractions based on the 0–0 peaks for solution samples in 2-methyl-tetrahydrofuran at 77 K.

**Figure 4 molecules-27-07981-f004:**
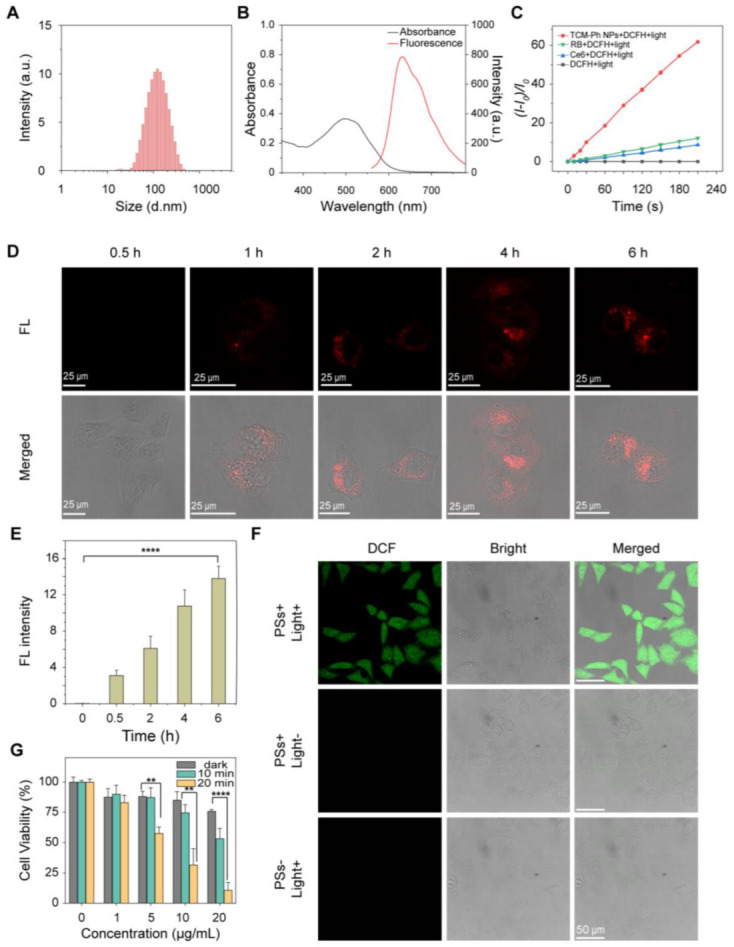
Efficient cell penetrating and tumor cell ablation of TCM-Ph NPs. (**A**) Size distribution of TCM-Ph NPs in water. (**B**) Absorbance and fluorescence spectra of TCM-Ph NPs in water, *λ*_ex_ = 497 nm. (**C**) Light-induced total ROS generation of Ce6, RB, and TCM-Ph NPs, evaluated by DCFH (40 μM). The plot of relative PL intensity (*I*-*I*_0_)/*I*_0_ at 525 nm versus the different irradiation times, where *I*_0_ is the fluorescence value of the mixture at 525 nm before illumination and *I* is the fluorescence value of the mixture at 525 nm after illumination, respectively, *λ*_ex_ = 488 nm. (**D**) CLSM of HeLa cells incubated with TCM-Ph NPs (10 μM based on TCM-Ph) for different times. Red channel from TCM-Ph NPs: *λ*_ex_ = 514 nm, *λ*_em_ = 550–700 nm. Scale bar: 25 μm. (**E**) Average fluorescence intensity of cells incubated with TCM-Ph NPs (10 μM based on TCM-Ph) for different times from Figure (**D**) (down). Data are shown as mean ± s.d., with *n* = 3, **** *p* < 0.0001. (**F**) Intracellular ROS level in HeLa cells under various treatments detected by reaction between DCFH and ROS. Green channel from DCFH: *λ*_ex_ = 488 nm, *λ*_em_ = 505–560 nm. Scale bar: 50 μm. (**G**) In vitro cytotoxicity of different concentrations of TCM-Ph NPs under dark, 10 min and 20 min white-light irradiation. Light: 50 mW cm^−2^, > 400 nm. Data are shown as mean ± s.d., with *n* = 3, **** *p* < 0.0001, ** *p* < 0.01.

## Data Availability

The data are available on request from the corresponding authors.

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
