# Peer review of "Tricyano-Methylene-Pyridine Based High-Performance Aggregation-Induced Emission Photosensitizer for Imaging and Photodynamic Therapy"

_molecules, 2022, doi:10.3390/molecules27227981_

Round 1

Reviewer 1 Report

Wang and co-workers developed novel TCM based AIE photosensitizer. And TCM-Ph with a strong donor-π-acceptor (D-π-A) structure, which greatly separated the HOMO-LUMO distribution and reduced the ΔEST, thereby increased the number of triplet excitons and produced more ROS. What’s more, bright near-infrared emission wavelength was given by this novel AIE structure. After co-packed with DSPE-PEG2000/DSPE-PEG2000-FA, NPs can be penetrated by Hela cells. A better PDT effect was displayed in cell experiment. This manuscript can provide reference for further research. Contents are proper for publication in Molecules after addressing the following problems.

Minor corrections:

1. Author should improve the resolution of each figures.

2. Figure 3D, DFT calculation showed bizarre charge separation for HOMO-LUMO of TCM-Et. Author can check calculation algorithm for opt. and freq. here.

3. In the part of DFT calculation, for excited states, author should provide more detailed energy levels from S1-S6 and T1-T6 in the main article. And I think the energy gap from S1 state to the nearest triple excited state was meaningful for the discussion of ΔEST as well.

4. As AIE PSs with better photophysical properties, I think the molar absorbance and brightness data should be given in the main article.

5. Some recently published papers in particular the review articles regarding to the AIE photosensitizers and photodynamic therapy (Aggregate 2021, 2, 95-113; Sci. China Chem. 2020, 63, 1428; Small Struct. 2022, 3, 2200036; JACS 2022, 144, 3429; Small Methods, 2020, 4, 2000013; 2020, 4, 2000046) could be considered to cite.

6. Figure 4E and 4G, author should provide statistical analysis of variances for experimental data.

Reviewer 2 Report

In this manuscript, Wang and coworkers reported an aggregation-induced emission (AIE) photosensitizer TCM-Ph, which showed bright near-infrared emission and high ROS generation capacity that can effectively eliminate cancer cells under image guidance. The results are interesting, however following issues should be addressed before acceptance.

1.     The authors have provided solid-state absolute fluorescence quantum yields of TCM-Et and TCM-Ph. For comparison, please also provide the absolute fluorescence quantum yields of the two compounds in THF solutions.

2.     Please provide TEM images of TCM-Ph NPs.

3.     The authors mentioned that after incubating for 6 h, the unpacked TCM-Ph could not be taken up by tumor cells, while the TCM-Ph NPs treatment cells exhibited efficient cell penetration with strong intracellular fluorescence. Please explain the reasons for this phenomenon.

4.     It will be better to have the in vivo experiments to further demonstrate the advantages of the prepared TCM-Ph.

Author Response

Please see the attacnment.

Reviewer 3 Report

This manuscript developed an AIE photosensitizer based on TCM with bright fluorescence and high ROS generation for cell imaging and photodynamic therapy. The molecular design is rational to improve the ROS generation ability. Cell killing effect was achieved using the TCM-Ph NPs under light irradiation. The whole manuscript was organized logically. I recommend the acceptance of this work after minor changes.

1. “TCM” appeared in the title at the first time without any explanations, which is not suitable.

2. The resolution of all the figures should be improved.

3. As important optical properties, the absorption spectra of TCM-Et, TCM-Ph, and TCM-Ph NPs were missing in the main text.

4. Are there other types of ROS production except for 1O2 for the developed TCM-Ph under light irradiation?

5. Figure 4D: the cell imaging is blurry. Where does the red fluorescence signal come from? The nuclear staining should be conducted to confirm the intracellular distribution of the nanoparticles. 
